# THE LOCAL ELASTICITY OF NEURAL NETWORKS

**Hangfeng He & Weijie J. Su**
University of Pennsylvania
Philadelphia, PA
`hangfeng@seas.upenn.edu, suw@wharton.upenn.edu`

## ABSTRACT

This paper presents a phenomenon in neural networks that we refer to as *local elasticity*. Roughly speaking, a classifier is said to be locally elastic if its prediction at a feature vector $x'$ is *not* significantly perturbed, after the classifier is updated via stochastic gradient descent at a (labeled) feature vector $x$ that is *dissimilar* to $x'$ in a certain sense. This phenomenon is shown to persist for neural networks with nonlinear activation functions through extensive simulations on real-life and synthetic datasets, whereas this is not observed in linear classifiers. In addition, we offer a geometric interpretation of local elasticity using the neural tangent kernel (Jacot et al., 2018). Building on top of local elasticity, we obtain pairwise similarity measures between feature vectors, which can be used for clustering in conjunction with $K$-means. The effectiveness of the clustering algorithm on the MNIST and CIFAR-10 datasets in turn corroborates the hypothesis of local elasticity of neural networks on real-life data. Finally, we discuss some implications of local elasticity to shed light on several intriguing aspects of deep neural networks.

## 1 INTRODUCTION

Neural networks have been widely used in various machine learning applications, achieving comparable or better performance than existing methods without requiring highly engineered features (Krizhevsky et al., 2012). However, neural networks have several intriguing aspects that defy conventional views of statistical learning theory and optimization, thereby hindering the architecture design and interpretation of these models. For example, despite having more parameters than training examples, deep neural networks generalize well without an explicit form of regularization (Zhang et al., 2017; Neyshabur et al., 2017; Arora et al., 2019a). Zhang et al. (2017) also observe that neural networks can perfectly fit corrupted labels while maintaining a certain amount of generalization power[1].

In this paper, we complement this line of findings by proposing a hypothesis that fundamentally distinguishes neural networks from linear classifiers[2]. This hypothesis is concerned with the dynamics of training neural networks using stochastic gradient descent (SGD). Indeed, the motivation is to address the following question:

> *How does the update of weights using SGD at an input $x$ and its label $y$*
> *impact the prediction of the neural networks at another input $x'$ ?*

Taking this dynamic perspective, we make the following three contributions.

First, we hypothesize that neural networks are *locally elastic* in the following sense: an extensive set of experiments on synthetic examples demonstrate that the impact on the prediction of $x'$ is significant if $x'$ is in a *local* vicinity of $x$ and the impact diminishes as $x'$ becomes far from $x$ in an *elastic* manner. In contrast, local elasticity is not observed in linear classifiers due to the leverage effect (Weisberg, 2005). Thus, at a high level, local elasticity must be inherently related

---

[1]This property also holds for the 1-nearest neighbors algorithm.
[2]Without using the kernel trick, these classifiers include linear regression, logistic regression, support vector machine, and linear neural networks.

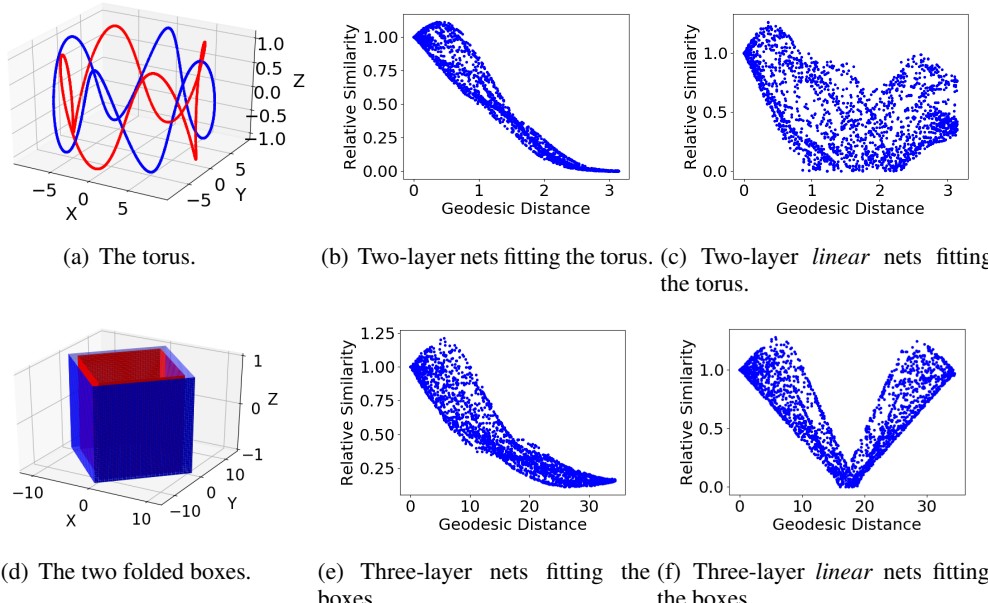

(a) The torus.  (b) Two-layer nets fitting the torus. (c) Two-layer *linear* nets fitting the torus.

(d) The two folded boxes.  (e) Three-layer nets fitting the (f) Three-layer *linear* nets fitting boxes.  the boxes.

Figure 1: Comparisons between ReLU neural networks and linear neural networks in terms of local elasticity. In the left column, the red points form one class and the blue points form the other class. The linear nets are of the same sizes as their non-linear counterparts. The details on how to construct the torus and boxes can be found in Appendix A.1 and the network architectures are described in Appendix A.4. During the training process of the neural networks, we plot the geodesic distance (see more details in Appendix A.1) between two blue points $x$ and $x'$, and their relative prediction changes (see its definition in Equation (2)) in (b), (c), (e), and (f). The correlations of (b), (c), (e), and (f) are $-0.97$, $-0.48$, $-0.92$, and $-0.14$, respectively. (b) and (e) show that the distance and the relative change exhibit decreasing monotonic relationship, thereby confirming local elasticity, while no monotonic relationship is found in (c) and (f).

to the nonlinearity of neural networks and SGD used in updating the weights. This phenomenon is illustrated by Figure 1. Additional synthetic examples and ImageNet (Deng et al., 2009) with a pre-trained ResNet (He et al., 2016) in Appendix A.2 further confirm local elasticity. For completeness, we remark that the notion of local elasticity seems related to influence functions on the surface (Koh & Liang, 2017). The fundamental distinction, however, is that the former takes into account the dynamics of the training process whereas the latter does not. See Section 2 for a formal introduction of the notion of local elasticity.

Furthermore, we devise a clustering algorithm by leveraging local elasticity of neural networks. In short, this algorithm records the relative change of the prediction on a feature vector to construct a similarity matrix of the training examples. Next, the similarity matrix is used by, for example, $K$-means to partition the points into different clusters. The experiments on MNIST (LeCun, 1998) and CIFAR-10 (Krizhevsky, 2009) demonstrate the effectiveness of this local elasticity-based clustering algorithm. For two superclasses (e.g., mammal and vehicle), the algorithm is capable of partitioning the mammal class into cat and dog, and the second superclass into car and truck. These empirical results, in turn, corroborate our hypothesis that neural networks (with nonlinear activation) are locally elastic. See the description of the algorithm in Section 3 and experimental results in Section 4.

Finally, this paper provides profound implications of local elasticity on memorization and generalization of neural networks, among others. In this spirit, this work seeks to shed light on some intriguing aspects of neural networks. Intuitively, the locality part of this property suggests that the neural networks can efficiently fit the label of an input without significantly affecting most examples that have been well fitted. This property is akin to the nearest neighbors algorithm (see, e.g., Papernot & McDaniel (2018)). Meanwhile, the elasticity part implies that the prediction surface is

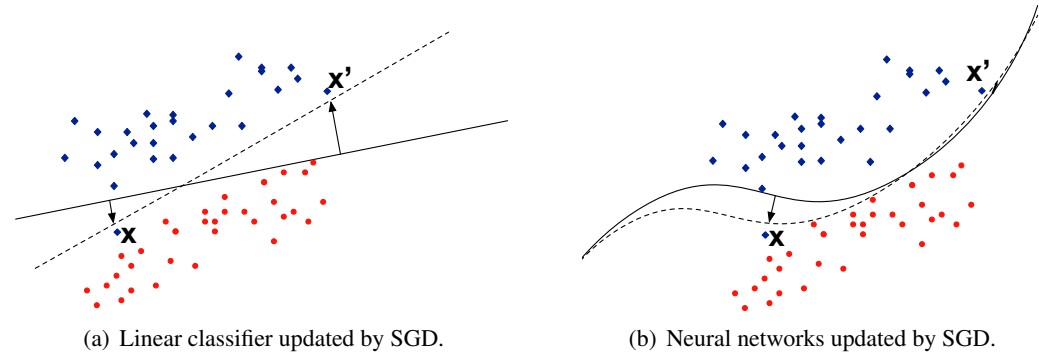

(a) Linear classifier updated by SGD.  (b) Neural networks updated by SGD.

Figure 2: An illustration of linear regression and neural networks updated via SGD. The prediction of $x'$ changes a lot after an SGD update on $x$ in the linear case, though $x'$ is far away from $x$. In contrast, the change in the prediction at $x'$ is rather small in the neural networks case.

likely to remain smooth in the training process, in effect regularizing the complexity of the nets in a certain sense. These implications are discussed in detail in Section 5.

## 1.1 RELATED WORK

There has been a line of work probing the geometric properties of the decision boundary of neural networks. Montufar et al. (2014) investigate the connection between the number of linear regions and the depth of a ReLU network and argue that a certain intrinsic rigidity of the linear regions may improve the generalization. Fawzi et al. (2017; 2018) observe that the learned decision boundary is flat along most directions for natural images. See Hanin & Rolnick (2019) for the latest development along this line and Fort et al. (2019) for a dynamic perspective on the landscape geometry.

In another related direction, much effort has been expended on the expressivity of neural networks, starting from universal approximation theorems for two-layer networks (Cybenko, 1989; Hornik et al., 1989; Barron, 1993). Lately, deep neural networks have been shown to possess better representational power than their shallow counterparts (Delalleau & Bengio, 2011; Telgarsky, 2016; Eldan & Shamir, 2016; Mhaskar & Poggio, 2016; Yarotsky, 2017; Chen et al., 2019). From a non-parametric viewpoint, approximation risks are obtained for neural networks under certain smooth assumptions on the regression functions (Schmidt-Hieber, 2017; Suzuki, 2018; Klusowski & Barron, 2018; Liang, 2018; Bauer & Kohler, 2019; E et al., 2019a).

A less related but more copious line of work focuses on optimization for training neural networks. A popular approach to tackling this problem is to study the optimization landscape of neural networks (Choromanska et al., 2015; Soudry & Hoffer, 2017; Zhou & Liang, 2017; Safran & Shamir, 2018; Du & Lee, 2018; Liang et al., 2018; Soltanolkotabi et al., 2018). Another approach is to analyze the dynamics of specific optimization algorithms applied to neural networks (Tian, 2017; Li & Yuan, 2017; Soltanolkotabi, 2017; Brutzkus & Globerson, 2017; Du et al., 2018; Li & Liang, 2018; Allen-Zhu et al., 2018; Zou et al., 2018; Du et al., 2019; Allen-Zhu et al., 2019). Alternatively, researchers have considered the evolution of gradient descent on two-layer neural networks using optimal transport theory (Song et al., 2018; Chizat & Bach, 2018; Sirignano & Spiliopoulos, 2019; Rotskoff & Vanden-Eijnden, 2018). More recently, there is a growing recognition of intimate similarities between over-parameterized neural networks and kernel methods from an optimization perspective (Zhang et al., 2017; Daniely, 2017; Belkin et al., 2018; Jacot et al., 2018; Yang, 2019; Arora et al., 2019b; Lee et al., 2019; E et al., 2019b). For completeness, some work demonstrates a certain superiority of neural networks in generalization over the corresponding kernel methods (Wei et al., 2018; Allen-Zhu & Li, 2019; Ghorbani et al., 2019).

## 2 LOCAL ELASTICITY

This section formalizes the notion of local elasticity and proposes associated similarity measures that will be used for clustering in Section 3. Denote by $\boldsymbol{x} \in \mathbb{R}^d$ and $y$ the feature vector and the label of an instance $(\boldsymbol{x}, y)$, respectively. Let $f(\boldsymbol{x}, \boldsymbol{w})$ be the prediction with model parameters $\boldsymbol{w}$, and write $\mathcal{L}(f, y)$ for the loss function. Consider using SGD to update the current parameters $\boldsymbol{w}$ using the instance $(\boldsymbol{x}, y)$:

$$\boldsymbol{w}^+ = \boldsymbol{w} - \eta \frac{\mathrm{d}\mathcal{L}(f(\boldsymbol{x}, \boldsymbol{w}), y)}{\mathrm{d}\boldsymbol{w}} = \boldsymbol{w} - \eta \frac{\partial \mathcal{L}(f(\boldsymbol{x}, \boldsymbol{w}), y)}{\partial f} \cdot \frac{\partial f(\boldsymbol{x}, \boldsymbol{w})}{\partial \boldsymbol{w}}. \tag{1}$$

In this context, we say that the classifier $f$ is *locally elastic* at parameters $\boldsymbol{w}$ if $|f(\boldsymbol{x}', \boldsymbol{w}^+) - f(\boldsymbol{x}', \boldsymbol{w})|$, the change in the prediction at a test feature vector $\boldsymbol{x}'$, is relatively *large* when $\boldsymbol{x}$ and $\boldsymbol{x}'$ are *similar/close*, and vice versa. Here, by similar/close, we mean two input points $\boldsymbol{x}$ and $\boldsymbol{x}'$ share many characteristics or are connected by a short geodesic path in the feature space. Intuitively, $\boldsymbol{x}$ and $\boldsymbol{x}'$ are similar if $\boldsymbol{x}$ denotes an Egyptian cat and $\boldsymbol{x}'$ denotes a Persian cat; they are dissimilar if $\boldsymbol{x}$ denotes a German shepherd and $\boldsymbol{x}'$ denotes a trailer truck.

For illustration, Figure 2(a) shows that the (linear) classifier is *not* locally elastic since the SGD update on $\boldsymbol{x}$ leads to significant impact on the prediction at $\boldsymbol{x}'$, though $\boldsymbol{x}'$ is far from $\boldsymbol{x}$; on the other hand, the (nonlinear) classifier in Figure 2(b) is locally elastic since the change at $\boldsymbol{x}'$ is relatively small compared with that at $\boldsymbol{x}$. In Section 2.3, we provide some intuition why nonlinearity matters for local elasticity in two-layer neural nets.

### 2.1 RELATIVE SIMILARITY

The essence of local elasticity is that the change in the prediction has an (approximate) monotonic relationship with the similarity of feature vectors. Therefore, the change can serve as a *proxy* for the similarity of two inputs $\boldsymbol{x}$ and $\boldsymbol{x}'$:

$$S_{\mathrm{rel}}(\boldsymbol{x}, \boldsymbol{x}') := \frac{|f(\boldsymbol{x}', \boldsymbol{w}^+) - f(\boldsymbol{x}', \boldsymbol{w})|}{|f(\boldsymbol{x}, \boldsymbol{w}^+) - f(\boldsymbol{x}, \boldsymbol{w})|}. \tag{2}$$

In the similarity measure above, $|f(\boldsymbol{x}, \boldsymbol{w}^+) - f(\boldsymbol{x}, \boldsymbol{w})|$ is the change in the prediction of $(\boldsymbol{x}, y)$ used for the SGD update, and $|f(\boldsymbol{x}', \boldsymbol{w}^+) - f(\boldsymbol{x}', \boldsymbol{w})|$ is the change in the output of a test input $\boldsymbol{x}'$. Note that $S_{\mathrm{rel}}(\boldsymbol{x}, \boldsymbol{x}')$ is not necessarily symmetric and, in particular, the definition depends on the weights $\boldsymbol{w}$. Our experiments in Section 4 suggest the use of a near-optimal $\boldsymbol{w}$ (which is also the case in the second definition of local elasticity in Section 2.2). In the notion of local elasticity, locality suggests that this ratio is large when $\boldsymbol{x}$ and $\boldsymbol{x}'$ are close, and vice versa, while elasticity means that this similarity measure decreases *gradually* and *smoothly*, as opposed to abruptly, when $\boldsymbol{x}'$ moves away from $\boldsymbol{x}$. Having evaluated this similarity measure for (almost) all pairs by performing SGD updates for several epochs, we can obtain a similarity matrix, whose rows are each regarded as a feature vector of its corresponding input. This can be used for clustering followed by $K$-means.

### 2.2 KERNELIZED SIMILARITY

Next, we introduce a different similarity measure that manifests local elasticity by making a connection to the neural tangent kernel (Jacot et al., 2018). Taking a small learning rate $\eta$, for a new feature point $\boldsymbol{x}'$, the change in its prediction due to the SGD update in Equation (1) approximately satisfies:

$$f(\boldsymbol{x}', \boldsymbol{w}^+) - f(\boldsymbol{x}', \boldsymbol{w}) = f\left(\boldsymbol{x}', \boldsymbol{w} - \eta \frac{\partial \mathcal{L}}{\partial f} \cdot \frac{\partial f(\boldsymbol{x}, \boldsymbol{w})}{\partial \boldsymbol{w}}\right) - f(\boldsymbol{x}', \boldsymbol{w})$$

$$\approx f(\boldsymbol{x}', \boldsymbol{w}) - \left\langle \frac{\partial f(\boldsymbol{x}', \boldsymbol{w})}{\partial \boldsymbol{w}}, \eta \frac{\partial \mathcal{L}}{\partial f} \cdot \frac{\partial f(\boldsymbol{x}, \boldsymbol{w})}{\partial \boldsymbol{w}} \right\rangle - f(\boldsymbol{x}', \boldsymbol{w})$$

$$= -\eta \frac{\partial \mathcal{L}}{\partial f} \left\langle \frac{\partial f(\boldsymbol{x}', \boldsymbol{w})}{\partial \boldsymbol{w}}, \frac{\partial f(\boldsymbol{x}, \boldsymbol{w})}{\partial \boldsymbol{w}} \right\rangle.$$

The factor $-\eta \frac{\partial \mathcal{L}}{\partial f}$ does not involve $\boldsymbol{x}'$, just as the denominator $|f(\boldsymbol{x}, \boldsymbol{w}^+) - f(\boldsymbol{x}, \boldsymbol{w})|$ in Equation (2). This observation motivates an alternative definition of the similarity:

$$S_{\mathrm{ker}}(\boldsymbol{x}, \boldsymbol{x}') := \frac{f(\boldsymbol{x}', \boldsymbol{w}) - f(\boldsymbol{x}', \boldsymbol{w}^+)}{\eta \frac{\partial \mathcal{L}(f(\boldsymbol{x}, \boldsymbol{w}), y)}{\partial f}}. \tag{3}$$

In the case of the $\ell_2$ loss $\mathcal{L}(f, y) = \frac{1}{2}(f - y)^2$, for example, $S_{\mathrm{ker}}(\boldsymbol{x}, \boldsymbol{x}') = \frac{f(\boldsymbol{x}', \boldsymbol{w}) - f(\boldsymbol{x}', \boldsymbol{w}^+)}{\eta(f - y)}$. In Section 3, we apply the kernel $K$-means algorithm to this similarity matrix for clustering.

The kernelized similarity in Equation (3) is approximately the inner product of the gradients of $f$ at $\boldsymbol{x}$ and $\boldsymbol{x}'$[3]. This is precisely the definition of the neural tangent kernel (Jacot et al., 2018) (see also Arora et al. (2019b)) if $\boldsymbol{w}$ is generated from i.i.d. normal distribution and the number of neurons in each layer tends to infinity. However, our empirical results suggest that a data-adaptive $\boldsymbol{w}$ may lead to more significant local elasticity. Explicitly, both similarity measures with pre-trained weights yield better performance of Algorithm 1 (in Section 3) than those with randomly initialized weights. This is akin to the recent findings on a certain superioritiy of data-adaptive kernels over their non-adaptive counterparts (Dou & Liang, 2019).

### 2.3 Interpretation via Two-layer Networks

We provide some intuition for the two similarity measures in Equation (2) and Equation (3) with two-layer neural networks. Letting $\overline{\boldsymbol{x}} = (\boldsymbol{x}^\top, 1)^\top \in \mathbb{R}^{d+1}$, denote the networks by $f(\boldsymbol{x}, \boldsymbol{w}) = \sum_{r=1}^{m} a_r \sigma(\boldsymbol{w}_r^\top \overline{\boldsymbol{x}})$, where $\sigma(\cdot)$ is the ReLU activation function ($\sigma(x) = x$ for $x \geq 0$ and $\sigma(x) = 0$ otherwise) and $\boldsymbol{w} = (\boldsymbol{w}_1^\top, \ldots, \boldsymbol{w}_r^\top)^\top \in \mathbb{R}^{m(d+1)}$. For simplicity, we set $a_k \in \{-1, 1\}$. Note that this does *not* affect the expressibility of this net due to the positive homogeneity of ReLU.

Assuming $\boldsymbol{w}$ are i.i.d. normal random variables and some other conditions, in the appendix we show that this neural networks with the SGD rule satisfies

$$\frac{f(\boldsymbol{x}', \boldsymbol{w}^+) - f(\boldsymbol{x}', \boldsymbol{w})}{f(\boldsymbol{x}, \boldsymbol{w}^+) - f(\boldsymbol{x}, \boldsymbol{w})} = (1 + o(1)) \frac{(\boldsymbol{x}^\top \boldsymbol{x}' + 1) \sum_{r=1}^{m} \mathbb{I}\{\boldsymbol{w}_r^T \overline{\boldsymbol{x}} \geq 0\} \mathbb{I}\{\boldsymbol{w}_r^\top \overline{\boldsymbol{x}}' \geq 0\}}{(\|\boldsymbol{x}\|^2 + 1) \sum_{r=1}^{m} \mathbb{I}\{\boldsymbol{w}_r^T \overline{\boldsymbol{x}} \geq 0\}}. \tag{4}$$

Above, $\|\cdot\|$ denotes the $\ell_2$ norm. For comparison, we get $\frac{\tilde{f}(\boldsymbol{x}', \boldsymbol{w}^+) - \tilde{f}(\boldsymbol{x}', \boldsymbol{w})}{\tilde{f}(\boldsymbol{x}, \boldsymbol{w}^+) - \tilde{f}(\boldsymbol{x}, \boldsymbol{w})} = \frac{\boldsymbol{x}^\top \boldsymbol{x}' + 1}{\|\boldsymbol{x}\|^2 + 1}$ for the linear neural networks $\tilde{f}(\boldsymbol{x}) := \sum_{r=1} a_r \boldsymbol{w}_r^\top \boldsymbol{x}$. For both cases, the change in the prediction at $\boldsymbol{x}'$ resulting from an SGD update at $(\boldsymbol{x}, y)$ involves a multiplicative factor of $\boldsymbol{x}^\top \boldsymbol{x}'$. However, the distinction is that the nonlinear classifier $f$ gives rise to a dependence of the signed $S_{\mathrm{ker}}(\boldsymbol{x}, \boldsymbol{x}')$ on the fraction of neurons that are activated at both $\boldsymbol{x}$ and $\boldsymbol{x}'$. Intuitively, a close similarity between $\boldsymbol{x}$ and $\boldsymbol{x}'$ can manifest itself with a large number of commonly activated neurons. This is made possible by the nonlinearity of the ReLU activation function. We leave the rigorous treatment of the discussion here for future work.

## 3 The Local Elasticity Algorithm for Clustering

This section introduces a novel algorithm for clustering that leverages the local elasticity of neural networks. We focus on the setting where all (primary) examples are from the same (known) super-class (e.g., mammal) and the interest is, however, to partition the primary examples into finer-grained (unknown) classes (e.g., cat and dog). To facilitate this process, we include an auxiliary dataset with all examples from a different superclass (e.g., vehicle). See Figure 3 for an illustration of the setting. To clear off any confusion, we remark that the aim is to corroborate the hypothesis of local elasticity by showing the effectiveness of this clustering algorithm.

The centerpiece of our algorithm is the dynamic construction of a matrix that records pairwise similarities between all primary examples. In brief, the algorithm operates as if it were learning to distinguish between the primary examples (e.g., mammals) and the auxiliary examples (e.g., vehicles) via SGD. On top of that, the algorithm evaluates the changes in the predictions for any pairs of primary examples during the training process, and the recorded changes are used to construct the pairwise similarity matrix based on either the relative similarity in Equation (2) or the kernelized similarity in Equation (3). Taking the mammal case as earlier, the rationale of the algorithm is that local elasticity is likely to yield a larger similarity score between two cats (or two dogs), and a smaller similarity score between a cat and a dog.

---

[3]As opposed to the relative similarity, this definition does not take absolute value in order to retain its interpretation as an inner product.

---

**Algorithm 1** The Local Elasticity Based Clustering Algorithm.

---

**Input:** primary dataset $\mathcal{P} = \{\boldsymbol{x}_i\}_{i=1}^n$, auxiliary dataset $\mathcal{A} = \{\widetilde{\boldsymbol{x}}_j\}_{j=1}^m$, classifier $f(\boldsymbol{x}, \boldsymbol{w})$, initial
weights $\boldsymbol{w}_0$, loss function $\mathcal{L}$, learning rate $\eta_t$, option $o \in \{\texttt{relative}, \texttt{kernelized}\}$

1: combine $\mathcal{D} = \{(\boldsymbol{x}_i, y_i = 1) \text{ for } \boldsymbol{x}_i \in \mathcal{P}\} \bigcup \{(\boldsymbol{x}_i, y_i = -1) \text{ for } \boldsymbol{x}_i \in \mathcal{A}\}$
2: set $S$ to $n \times n$ matrix of all zeros
3: **for** $t = 1$ to $n + m$ **do**
4:     sample $(\boldsymbol{x}, y)$ from $\mathcal{D}$ w/o replacement
5:     $\boldsymbol{w_t} = \texttt{SGD}(\boldsymbol{w}_{t-1}, \boldsymbol{x}, y, f, \mathcal{L}, \eta_t)$
6:     **if** $y = 1$ **then**
7:         $\boldsymbol{p}_t = \texttt{Predict}(\boldsymbol{w}_t, \mathcal{P}, f)$
8:         find $1 \le i \le n$ such that $\boldsymbol{x} = \boldsymbol{x}_i \in \mathcal{P}$
9:         **if** $o = \texttt{relative}$ **then**
10:             $\boldsymbol{s}_t = \frac{|\boldsymbol{p}_t - \boldsymbol{p}_{t-1}|}{|\boldsymbol{p}_t(i) - \boldsymbol{p}_{t-1}(i)|}$
11:         **else**
12:             $g_t = \texttt{GetGradient}(\boldsymbol{w}_{t-1}, \boldsymbol{x}, y, f, \mathcal{L})$
13:             $\boldsymbol{s}_t = \frac{\boldsymbol{p}_t - \boldsymbol{p}_{t-1}}{-\eta_t \times g_t}$
14:         **end if**
15:     **end if**
16:     set the $i$th row $S(i, :) = \boldsymbol{s}_t$
17: **end for**
18: $S_{\text{symm}} = \frac{1}{2}(S + S^\top)$
19: $\boldsymbol{y}_{\text{subclass}} = \text{Clustering}(S_{\text{symm}})$
20: **return** $\boldsymbol{y}_{\text{subclass}}$

---

This method is formally presented in Algorithm 1, with elaboration as follows. While the initial parameters $\boldsymbol{w}_0$ are often set to i.i.d. random variables, our experiments suggest that "pre-trained" weights can lead to better clustering performance. Hence, we use a warm-up period to get nearly optimal $\boldsymbol{w}_0$ by training on the combined dataset $\mathcal{D}$. When the SGD is performed at a primary example $\boldsymbol{x}_i$ (labeled $y_i = 1$) during the iterations, the function $\texttt{Predict}(\boldsymbol{w}_i, \mathcal{P}, f) \in \mathbb{R}^n$ evaluates the predictions for all primary examples at $\boldsymbol{w}_i$, and these results are used to compute similarity measures between $\boldsymbol{x}_i$ and other primary feature vectors using Equation (2) or Equation (3), depending on the option. If one chooses to use the kernelized similarity, the function $\texttt{GetGradient}$ is called to the gradient of the loss function $\mathcal{L}$ at $\boldsymbol{w}_{t-1}, (\boldsymbol{x}, y)$ with respect to $f$. In practice, we can repeat the loop multiple times and then average the similarity matrices over the epochs. Finally, using the symmetrized $S_{\text{symm}}$, we apply an off-the-shelf clustering method such as $K$-means (for relative similarity) or kernel $K$-means (for kernel similarity) to partition the primary dataset $\mathcal{P}$ into subclasses.

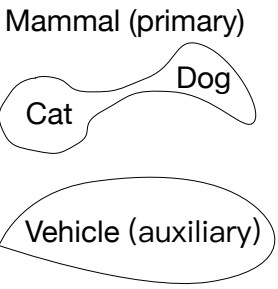

Figure 3: Illustration of the primary dataset and auxiliary dataset taken as input by the local elasticity based clustering algorithm.

## 4 EXPERIMENTS

### 4.1 EXPERIMENTAL SETTINGS

We evaluate the performance of Algorithm 1 on MNIST (LeCun, 1998) and CIFAR-10 (Krizhevsky, 2009). For the MNIST dataset, we choose the 6 pairs of digits that are the most difficult for the binary $K$-means clustering. Likewise, 6 pairs of classes are selected from CIFAR-10. For each pair (e.g., 5 and 8), we construct the primary data set by randomly sampling a total of 1000 examples equally from the two classes in the pair. The auxiliary dataset consists of 1000 examples that are randomly drawn from one or two different classes (in the case of two classes, evenly distribute the 1000 examples across the two classes).

Our experiments consider two-layer feedforward neural networks (FNN, which is also used in Figure 1), CNN (Krizhevsky et al., 2012), and ResNet (He et al., 2016). We use 40960 neurons with the ReLU activation function for two-layer FNN and details of the other architectures can be found

| Primary Examples | 5 vs 8 | 4 vs 9 | 7 vs 9 | 5 vs 9 | 3 vs 5 | 3 vs 8 |
|---|---|---|---|---|---|---|
| Auxiliary Examples | 3, 9 | 5, 7 | 4, 5 | 4, 8 | 8, 9 | 5 |
| $K$-means | 50.4 | 54.5 | 55.5 | 56.3 | 69.0 | 76.5 |
| PCA + $K$-means | 50.4 | 54.5 | 55.7 | 56.5 | 70.7 | 76.4 |
| $\ell_2$-relative(linear) | 51.0 | 54.6 | 51.3 | 58.8 | 58.3 | 58.7 |
| $\ell_2$-kernelized (linear) | 50.1 | 55.5 | 55.5 | 56.3 | 69.3 | 76.1 |
| BCE-relative (linear) | 50.1 | 50.0 | 50.2 | 50.6 | 51.4 | 50.1 |
| BCE-kernelized (linear) | 50.7 | 56.7 | 62.3 | 55.2 | 53.9 | 51.6 |
| $\ell_2$-relative (ours) | **75.9** | 55.6 | 62.5 | **89.3** | 50.3 | 74.7 |
| $\ell_2$-kernelized (ours) | 71.0 | **63.8** | **64.6** | 67.8 | **71.5** | **78.8** |
| BCE-relative (ours) | 50.2 | 50.5 | 51.9 | 55.7 | 53.7 | 50.2 |
| BCE-kernelized (ours) | 52.1 | 59.3 | 64.5 | 58.2 | 52.5 | 51.8 |

Table 1: Classification accuracy of Algorithm 1 and other methods on the MNIST dataset. BCE stands for the binary cross-entropy loss. We use $K$-means for relative similarity based clustering algorithms, and kernel $K$-means for kernelized similarity based clustering algorithms. For example, BCE-kernelized (linear) denotes the setting that uses the BCE loss, kernelized similarity and the linear activation function. The highest accuracy score in each study is in boldface.

| Primary Examples | Car vs Cat | Car vs Horse | Cat vs Bird | Dog vs Deer | Car vs Bird | Deer vs Frog |
|---|---|---|---|---|---|---|
| Auxiliary Examples | Bird | Cat, Bird | Car | Frog | Cat | Dog |
| $K$-means | 50.3 | 50.9 | 51.1 | 51.6 | 51.8 | 52.4 |
| PCA + $K$-means | 50.6 | 51.0 | 51.1 | 51.6 | 51.7 | 52.4 |
| $\ell_2$-relative (linear) | 50.0 | 56.8 | 55.2 | 50.6 | 56.3 | 52.5 |
| $\ell_2$-kernelized (linear) | 50.4 | 50.7 | 51.1 | 51.3 | 51.9 | 52.8 |
| BCE-relative (linear) | 50.4 | 50.9 | 50.2 | 50.6 | 55.3 | 50.2 |
| BCE-kernelized (linear) | 55.7 | 58.1 | 52.0 | 50.5 | 55.8 | 53.0 |
| $\ell_2$-relative (ours) | 53.7 | 50.2 | **58.4** | **54.2** | **63.0** | **53.5** |
| $\ell_2$-kernelized (ours) | 52.1 | 50.5 | 53.6 | 52.7 | 50.5 | 53.0 |
| BCE-relative (ours) | 51.1 | 53.5 | 50.7 | 51.1 | 55.6 | 50.3 |
| BCE-kernelized (ours) | **58.4** | **59.7** | 56.6 | 51.2 | 55.0 | 51.8 |

Table 2: Classification accuracy of Algorithm 1 and other methods on the CIFAR-10 dataset.

in Appendix A.4. We consider two types of loss functions, the $\ell_2$ loss and the cross-entropy loss $\mathcal{L}(f, y) = -y \log(f)$. Note that neural networks with the cross-entropy loss have an extra sigmoid layer on top of all layers. We run Algorithm 1 in one epoch. For comparison, linear neural networks (with the identity activation function) with the same architectures of their nonlinear counterparts are used as baseline models. Here, the use of linear neural networks instead of linear regression is to prevent possible influence of over-parameterization on our experimental results[4]. We also consider $K$-means and principal component analysis (PCA) followed by $K$-means as simple baselines.

## 4.2 RESULTS

**Comparison between relative and kernelized similarities.** Table 1 and Table 2 display the results on MNIST and CIFAR-10, respectively. The relative similarity based method with the $\ell_2$ loss performs best on CIFAR-10, while the kernelized similarity based method with the $\ell_2$ loss outperforms the other methods on MNIST. Overall, our methods outperform both linear and simple baseline models. These results demonstrate the effectiveness of Algorithm 1 and confirm our hypothesis of local elasticity in neural nets as well.

**Comparison between architectures.** The results of Algorithm 1 with the aforementioned three types of neural networks, namely FNN, CNN, and ResNet on MNIST are presented in Table 3. The results show that CNN in conjunction with the kernelized similarity has high classification accuracy. In contrast, the simple three-layer ResNet seems to not capture local elasticity.

---

[4]In fact, our experiments show that the two models exhibit very similar behaviors.

| Primary Examples | 5 vs 8 | 4 vs 9 | 7 vs 9 | 5 vs 9 | 3 vs 5 | 3 vs 8 |
|---|---|---|---|---|---|---|
| Auxiliary Examples | 3, 9 | 5, 7 | 4, 5 | 4, 8 | 8, 9 | 5 |
| $\ell_2$-relative (FNN) | **75.9** | 55.6 | 62.5 | 89.3 | 50.3 | 74.7 |
| $\ell_2$-kernelized (FNN) | 71.0 | 63.8 | 64.6 | 67.8 | 71.5 | 78.8 |
| $\ell_2$-relative (CNN) | 54.2 | 53.7 | 89.1 | 50.1 | 50.1 | 83.0 |
| $\ell_2$-kernelized (CNN) | 64.1 | **69.5** | **91.3** | **97.6** | **75.3** | **87.4** |
| $\ell_2$-relative (ResNet) | 50.7 | 55.0 | 55.5 | 78.3 | 52.3 | 52.3 |
| $\ell_2$-kernelized (ResNet) | 50.2 | 60.4 | 54.8 | 76.3 | 66.9 | 68.8 |

Table 3: Comparison of different architectures for the local elasticity based clustering algorithm. We only consider the $\ell_2$ loss on MNIST here for the sake of simplicity.

| Primary Examples | 5 vs 8 | 4 vs 9 | 7 vs 9 | 5 vs 9 | 3 vs 5 | 3 vs 8 |
|---|---|---|---|---|---|---|
| Auxiliary Examples | 3, 9 | 5, 7 | 4, 5 | 4, 8 | 8, 9 | 5 |
| AutoEncoder | 50.9 | 54.0 | 61.8 | 70.0 | 64.3 | 67.1 |
| ResNet-152 | **85.2** | **94.2** | **93.1** | 82.0 | 65.8 | **96.0** |
| $\ell_2$-relative (ours) | 54.2 | 53.7 | 89.1 | 50.1 | 50.1 | 83.0 |
| $\ell_2$-kernelized (ours) | 64.1 | 69.5 | 91.3 | **97.6** | **75.3** | 87.4 |

Table 4: Comparison with other feature extraction methods. For simplicity, we only consider CNN with $\ell_2$ loss on MNIST. The features from autoencoder and ResNet-152 are clustered by $K$-means.

To further evaluate the effectiveness of the local elasticity based Algorithm 1, we compare this clustering algorithm using CNN with two types of feature extraction approaches: autoencoder[5] and pre-trained ResNet-152 (He et al., 2016). An autoencoder reconstructs the input data in order to learn its hidden representation. ResNet-152 is pre-trained on ImageNet (Deng et al., 2009) and can be used to extract features of images. The performance of those models on MNIST[6] are shown in Table 4. Overall, autoencoder yields worst results. Although ResNet-152 performs quite well in general, yet our methods outperform it in some cases. It is an interesting direction to combine the strength of our algorithm and ResNet-152 in classification tasks that are very different from ImageNet. Moreover, unlike ResNet-152, our methods do not require a large number of examples for pre-training[7]. To better appreciate local elasticity and Algorithm 1, we study the effect of parameter initialization, auxiliary examples and normalized kernelized similarity in Appendix A.4. More discussions on the expected relative change, activation patterns, and activation functions can also be found in Appendix A.4.

## 5 IMPLICATIONS AND FUTURE WORK

In this paper, we have introduced a notion of local elasticity for neural networks. This notion enables us to develop a new clustering algorithm, and its effectiveness on the MNIST and CIFAR-10 datasets provide evidence in support of the local elasticity phenomenon in neural networks. While having shown the local elasticity in both synthetic and real-world datasets, we acknowledge that a mathematical foundation of this notion is yet to be developed. Specifically, how to rigorous formulate this notion for neural networks? A good solution to this question would involve the definition of a meaningful similarity measure and, presumably, would need to reconcile the notion with possible situations where the dependence between the prediction change and the similarity is not necessarily monotonic. Next, can we prove that this phenomenon occurs under some geometric structures of the dataset? Notably, recent evidence suggests that the structure of the training data has a profound im-

---

[5]We use the autoencoder in `https://github.com/L1aoXingyu/pytorch-beginner/blob/master/08-AutoEncoder/conv_autoencoder.py`.

[6]Here, the comparison between our methods and ResNet-152 on CIFAR-10 is not fair since ResNet-152 is trained using samples from the same classes as CIFAR-10.

[7]As a remark, here we do not compare with other clustering methods that are based on neural networks, such as DEC (Xie et al., 2016). We can replace $K$-means by other clustering algorithms to possibly improve the performance in some situations.

pact on the performance of neural networks (Goldt et al., 2019). Moreover, how does local elasticity depend on network architectures, activation functions, and optimization strategies?

Broadly speaking, local elasticity implies that neural networks can be plausibly thought of as a local method. We say a method is local if it seeks to fit a data point only using observations within a *window* of the data point. Important examples include the $k$-nearest neighbors algorithm, kernel smoothing, local polynomial regression (Fan, 2018), and locally linear embedding (Roweis & Saul, 2000). An exciting research direction is to formally relate neural networks to local methods. As opposed to the aforementioned classic local methods, however, neural networks seem to be capable of choosing the right *bandwidth* (window size) by adapting the data structure. It would be of great interest to show whether or not this adaptivity is a consequence of the elasticity of neural networks.

In closing, we provide further implications of this notion by seeking to interpret various aspects of neural networks via local elasticity. Our discussion lacks rigor and hence much future investigation is needed.

**Memorization.** Neural networks are empirically observed to be capable of fitting even random labels perfectly (Zhang et al., 2017), with provable guarantees under certain conditions (Allen-Zhu et al., 2019; Du et al., 2019; Oymak & Soltanolkotabi, 2019). Intuitively, local elasticity leads neural nets to progressively fit the examples in a vicinity of the input via each SGD update, while remaining fitting (most) labels that have been learned previously. A promising direction for future work is to relate local elasticity to memorization in a more concrete fashion.

**Stability and generalization.** Bousquet & Elisseeff (2002) demonstrate that the uniform stability of an algorithm implies generalization on a test dataset. As noted by Kuzborskij & Lampert (2018), due to its distribution-free and worst-case nature, uniform stability can lead to very loose bounds on generalization. The local elasticity of neural networks, however, suggests that the replace of one training point by another imposes limited perturbations on the predictions of most points and thus the loss might be more stable than expected. In this regard, it is possible to introduce a new notion of stability that relies on the metric of the input space for better generalization bounds.

**Data normalization and batch normalization.** The two normalization techniques are commonly used to improve the performance of neural networks (Gonzalez & Woods, 2002; Ioffe & Szegedy, 2015). The local elasticity viewpoint appears to imply that these techniques allow for a more solid relationship between the relative prediction change and a certain distance between feature vectors. Precisely, writing the lifting map $\varphi(\boldsymbol{x}) \approx \frac{\partial f(\boldsymbol{x}, \boldsymbol{w})}{\partial \boldsymbol{w}}$, Section 2.2 reveals that the kernelized similarity approximately satisfies $2S_{\mathrm{ker}}(\boldsymbol{x}, \boldsymbol{x}') \approx \|\varphi(\boldsymbol{x})\|^2 + \|\varphi(\boldsymbol{x}')\|^2 - d(\boldsymbol{x}, \boldsymbol{x}')^2$, where $d(\boldsymbol{x}, \boldsymbol{x}') = \|\varphi(\boldsymbol{x}) - \varphi(\boldsymbol{x}')\|$. To obtain a negative correlation between the similarity and the distance, therefore, one possibility is to have the norms of $\varphi(\boldsymbol{x})$ and $\varphi(\boldsymbol{x}')$ about equal to a constant. This might be made possible by employing the normalization techniques. See some empirical results in Appendix A.4. A venue for future investigation is to consolidate this implication on normalization techniques.

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

## A  EXPERIMENTAL DETAILS AND ADDITIONAL RESULTS

### A.1  CONSTRUCTION OF THE SYNTHETIC EXAMPLES IN FIGURE 1

The blue examples in the torus function (Figure 1(a)) can be defined parametrically by:

$$
\begin{aligned}
x(\theta) &= (8 + \sin(4\theta)) \cos\theta \\
y(\theta) &= (8 + \sin(4\theta)) \sin\theta \\
z(\theta) &= \cos(4\theta)
\end{aligned}
\tag{5}
$$

Similarly, the red examples in the torus function are defined by:

$$
\begin{aligned}
x(\theta) &= (8 - \sin(4\theta)) \cos\theta \\
y(\theta) &= (8 - \sin(4\theta)) \sin\theta \\
z(\theta) &= -\cos(4\theta)
\end{aligned}
\tag{6}
$$

The blue examples in the two folded boxes function (Figure 1(d)) are sampled from:

$$
\begin{aligned}
|y| + 1.2\,|x| &= 13 \\
z &\in [-1, 1]
\end{aligned}
\tag{7}
$$

Similarly, the red examples in the two folded boxes function are defined as:

$$
\begin{aligned}
|y| + 1.2\,|x| &= 11 \\
z &\in [-1, 1]
\end{aligned}
\tag{8}
$$

Geodesic distance is used to measure the shortest path between two points in a surface, or more generally in a Riemannian manifold. For example, in Figure 1, the geodesic distance for the torus function and the two folded boxes function is the distance in the curve and the distance in the surface rather than the Euclidean distance.

### A.2  MORE SIMULATIONS

More simulations can be found in Figure 4.

We further explore the local elasticity of ResNet-152 on ImageNet. We find that, when the pre-trained ResNet-152 are updated on a tabby cat via SGD, the predictions of tiger cats change more drastically than the predictions of warplanes. We randomly pick $50$ examples from the tabby cat synset as updating points, $25$ examples from the tiger cat synset and $25$ examples from the warplane synset as testing points. After updating on a tabby cat, we can get the average change of predictions on the $25$ tiger cats and that on the $25$ warplanes. By conducting a Wilcoxon rank-sum test with SGD updates on $50$ different tabby cats, we find that the average change of the predictions on the tiger cat are significantly more drastic than that on the warplane with a p-value of $0.0007$. The overall average relative similarity between the tiger cats and the tabby cats is $4.4$, while the overall average similarity between the warplanes and the tabby cats is $6.6$. Note that the relative similarity is computed from the relative KL divergence between the original prediction and the updated prediction.

Specifically, we find that the change of the predictions (KL divergence of $0.03$) on the tiger cat (Figure 5(b)) is more drastic than that (KL divergence of $0.002$) on the warplane (Figure 5(c)) after a SGD update on the tabby cat (Figure 5(a)), although the Euclidean distance ($511$) between the warplane and the tabby cat is smaller than that ($854$) between the tiger cat and the tabby cat.

Moreover, we conduct a simulation study to examine local elasticity in the case of using mini-batch SGD for updates. Specifically, we consider updating the weights of ResNet-152 in one iteration using two images. The training and test images are displayed in Figure 6. We find that the changes of the predictions on the tabby cat (Figure 6(c), KL divergence of $0.021$) and on the warplane (Figure 6(d), KL divergence of $0.024$) are more substantial than that on the tree frog (Figure 6(e), KL divergence of $0.0004$) after an SGD update on the minibatch composed of a tabby cat (Figure 6(a)) and a warplane (Figure 6(b)).

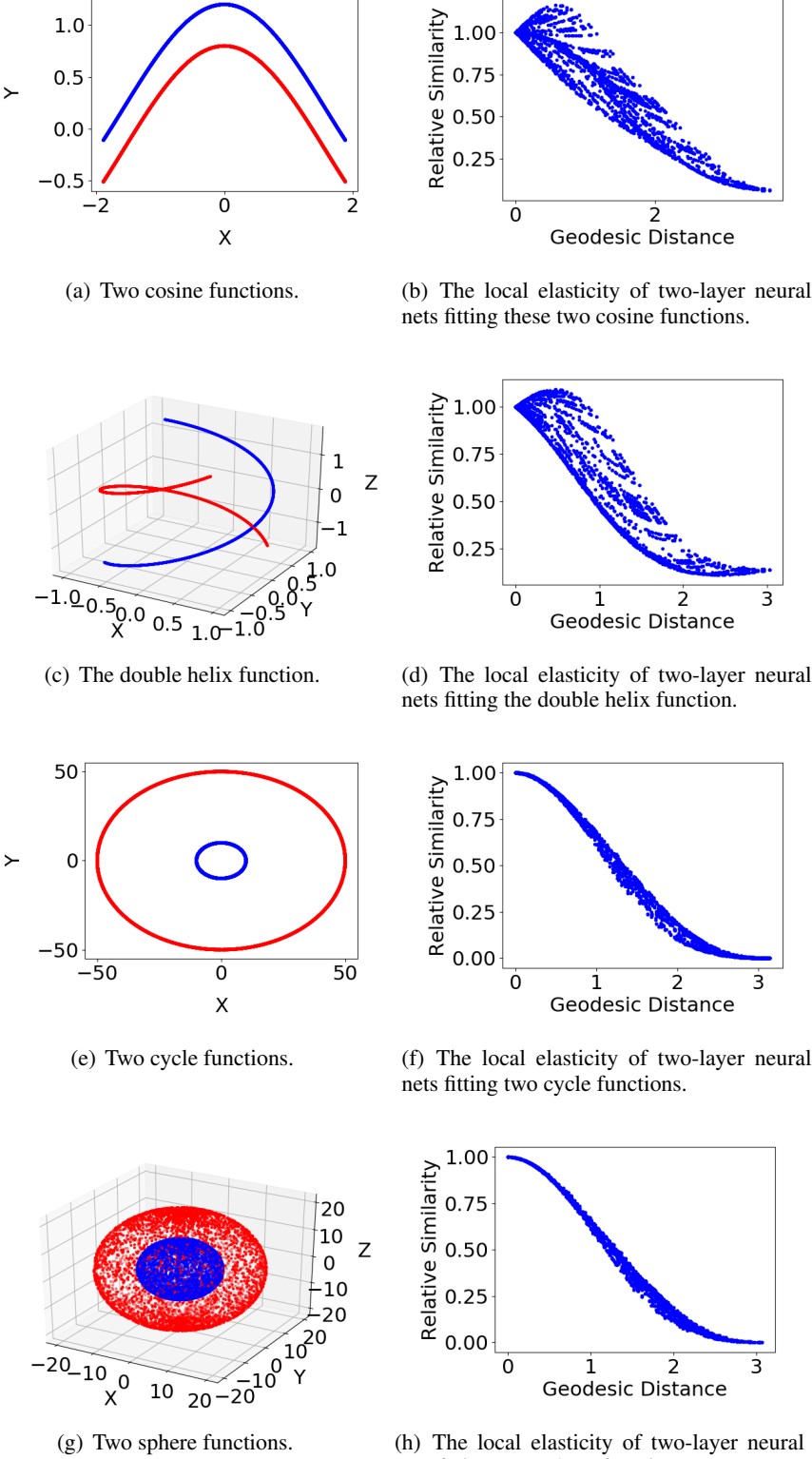

(a) Two cosine functions.

(b) The local elasticity of two-layer neural nets fitting these two cosine functions.

(c) The double helix function.

(d) The local elasticity of two-layer neural nets fitting the double helix function.

(e) Two cycle functions.

(f) The local elasticity of two-layer neural nets fitting two cycle functions.

(g) Two sphere functions.

(h) The local elasticity of two-layer neural nets fitting two sphere functions.

Figure 4: More simulations of the local elasticity.

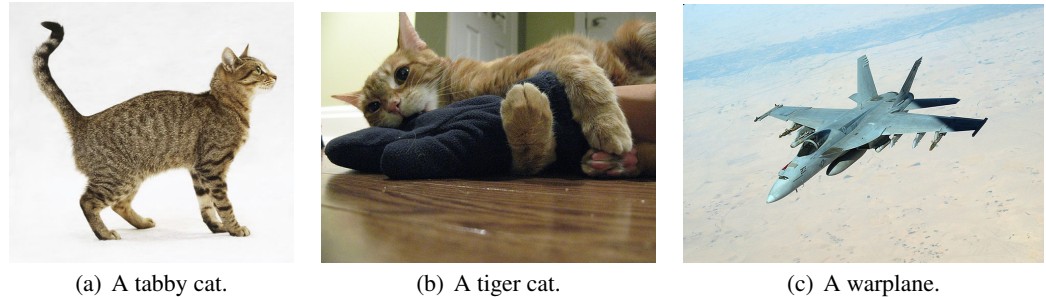

(a) A tabby cat.    (b) A tiger cat.    (c) A warplane.

Figure 5: Simulations in real data. We show specific examples for a tabby cat, a tiger cat and a warplane.

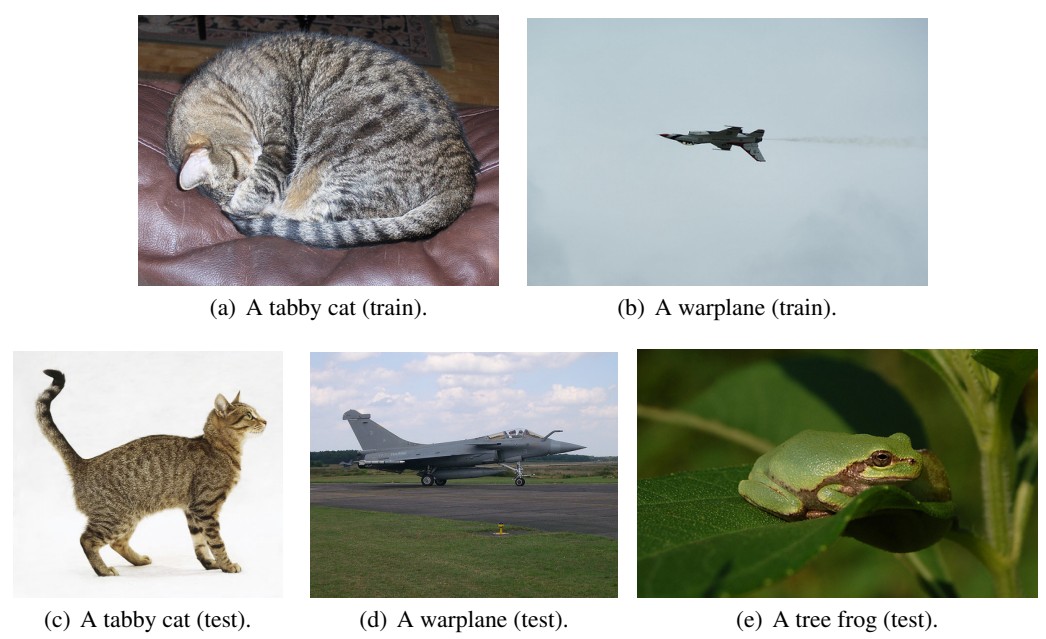

(a) A tabby cat (train).    (b) A warplane (train).

(c) A tabby cat (test).    (d) A warplane (test).    (e) A tree frog (test).

Figure 6: Simulations showing local elasticity with mini-batch SGD.

### A.3 SOME CALCULATIONS

We brief explain how to obtain Equation (4) under the same assumptions of Li & Liang (2018).

$$
\begin{aligned}
f(\overline{\boldsymbol{x}}', \boldsymbol{w}^+) - f(\boldsymbol{x}', \boldsymbol{w}) &= \sum_{r=1}^{m} a_r \sigma((\boldsymbol{w}_r^+)^\top \boldsymbol{x}') - \sum_{r=1}^{m} a_r \sigma(\boldsymbol{w}_r^\top \boldsymbol{x}') \\
&= \sum_{r=1}^{m} a_r \sigma\left(\left(\boldsymbol{w}_r - \eta \frac{\partial \mathcal{L}}{\partial f} a_r \boldsymbol{x} \mathbb{I}\{\boldsymbol{w}_r^T \boldsymbol{x} \geq 0\}\right)^\top \boldsymbol{x}'\right) - \sum_{r=1}^{m} a_r \sigma(\boldsymbol{w}_r^\top \boldsymbol{x}') \\
&= \sum_{r=1}^{m} a_r \sigma\left(\boldsymbol{w}_r^\top \boldsymbol{x}' - \eta \frac{\partial \mathcal{L}}{\partial f} a_r \boldsymbol{x}^\top \boldsymbol{x}' \mathbb{I}\{\boldsymbol{w}_r^T \boldsymbol{x} \geq 0\}\right) - \sum_{r=1}^{m} a_r \sigma(\boldsymbol{w}_r^\top \boldsymbol{x}') \\
&= \Delta_1 + \Delta_2,
\end{aligned}
$$

where

$$\Delta_1 = -\eta \frac{\partial \mathcal{L}}{\partial f} \sum_{r=1}^{m} \boldsymbol{x}^\top \boldsymbol{x}' \mathbb{I}\{\boldsymbol{w}_r^T \boldsymbol{x} \geq 0\} \mathbb{I}\{\boldsymbol{w}_r^\top \boldsymbol{x}' \geq 0\}$$

$$= -\eta \frac{\partial \mathcal{L}}{\partial f} \boldsymbol{x}^\top \boldsymbol{x}' \sum_{r=1}^{m} \mathbb{I}\{\boldsymbol{w}_r^T \boldsymbol{x} \geq 0\} \mathbb{I}\{\boldsymbol{w}_r^\top \boldsymbol{x}' \geq 0\}$$

and

$$\|\Delta_2\| \leq \sum_{r=1}^{m} \left| -a_r \eta \frac{\partial \mathcal{L}}{\partial f} a_r \boldsymbol{x}^\top \boldsymbol{x}' \mathbb{I}\{\boldsymbol{w}_r^T \boldsymbol{x} \geq 0\} \right| \cdot \left| \mathbb{I}\left\{ \boldsymbol{w}_r^\top \boldsymbol{x}' - \eta \frac{\partial \mathcal{L}}{\partial f} a_r \boldsymbol{x}^\top \boldsymbol{x}' \mathbb{I}\{\boldsymbol{w}_r^T \boldsymbol{x} \geq 0\} \geq 0 \right\} - \mathbb{I}\{\boldsymbol{w}_r^\top \boldsymbol{x}' \geq 0\} \right|$$

$$= \eta \left| \frac{\partial \mathcal{L}}{\partial f} \right| |\boldsymbol{x}^\top \boldsymbol{x}'| \sum_{r=1}^{m} \mathbb{I}\{\boldsymbol{w}_r^T \boldsymbol{x} \geq 0\} \left| \mathbb{I}\left\{ \boldsymbol{w}_r^\top \boldsymbol{x}' - \eta \frac{\partial \mathcal{L}}{\partial f} a_r \boldsymbol{x}^\top \boldsymbol{x}' \mathbb{I}\{\boldsymbol{w}_r^T \boldsymbol{x} \geq 0\} \geq 0 \right\} - \mathbb{I}\{\boldsymbol{w}_r^\top \boldsymbol{x}' \geq 0\} \right|$$

$$= \eta \left| \frac{\partial \mathcal{L}}{\partial f} \right| |\boldsymbol{x}^\top \boldsymbol{x}'| \sum_{r=1}^{m} \mathbb{I}\{\boldsymbol{w}_r^T \boldsymbol{x} \geq 0\} \left| \mathbb{I}\left\{ \boldsymbol{w}_r^\top \boldsymbol{x}' - \eta \frac{\partial \mathcal{L}}{\partial f} a_r \boldsymbol{x}^\top \boldsymbol{x}' \geq 0 \right\} - \mathbb{I}\{\boldsymbol{w}_r^\top \boldsymbol{x}' \geq 0\} \right|.$$

As shown in Li & Liang (2018), only a vanishing fraction of the neurons lead to different patterns between $\mathbb{I}\left\{ \boldsymbol{w}_r^\top \boldsymbol{x}' - \eta \frac{\partial \mathcal{L}}{\partial f} a_r \boldsymbol{x}^\top \boldsymbol{x}' \geq 0 \right\}$ and $\mathbb{I}\{\boldsymbol{w}_r^\top \boldsymbol{x}' \geq 0\}$. Consequently, we get $|\Delta_2| \ll |\Delta_1|$, thereby certifying

$$f(\boldsymbol{x}', \boldsymbol{w}^+) - f(\boldsymbol{x}', \boldsymbol{w}) = -(1 + o(1))\eta \frac{\partial \mathcal{L}}{\partial f} \boldsymbol{x}^\top \boldsymbol{x}' \sum_{r=1}^{m} \mathbb{I}\{\boldsymbol{w}_r^T \boldsymbol{x} \geq 0\} \mathbb{I}\{\boldsymbol{w}_r^\top \boldsymbol{x}' \geq 0\}$$

$$= (1 + o(1)) \frac{\boldsymbol{x}^\top \boldsymbol{x}' \sum_{r=1}^{m} \mathbb{I}\{\boldsymbol{w}_r^T \boldsymbol{x} \geq 0\} \mathbb{I}\{\boldsymbol{w}_r^\top \boldsymbol{x}' \geq 0\}}{\|\boldsymbol{x}\|^2 \sum_{r=1}^{m} \mathbb{I}\{\boldsymbol{w}_r^T \boldsymbol{x} \geq 0\}} (f(\boldsymbol{x}, \boldsymbol{w}^+) - f(\boldsymbol{x}, \boldsymbol{w})).$$

## A.4 MORE ASPECTS ON THE EXPERIMENTS

| Primary Examples | 5 vs 8 | 4 vs 9 | 7 vs 9 | 5 vs 9 | 3 vs 5 | 3 vs 8 |
|---|---|---|---|---|---|---|
| Auxiliary Examples | 3, 9 | 5, 7 | 4, 5 | 4, 8 | 8, 9 | 5 |
| $\ell_2$-relative (random) | 50.9 | 55.2 | 51.2 | 59.0 | 58.5 | 59.9 |
| $\ell_2$-kernelized (random) | 50.4 | 55.5 | 55.7 | 56.6 | 68.4 | 75.9 |
| $\ell_2$-relative (ours) | **75.9** | 55.6 | 62.5 | **89.3** | 50.3 | 74.7 |
| $\ell_2$-kernelized (ours) | 71.0 | **63.8** | **64.6** | 67.8 | **71.5** | **78.8** |

Table 5: Comparison of two different settings, random and optimal (default), for parameter initialization. For simplicity, we only consider $\ell_2$ loss on MNIST here.

**Architectures.** We use 40960 neurons with ReLU activation function for two-layer neural nets in simulations and experiments. We use 8192 neurons for each hidden layer with ReLU activation function in three-layer neural networks in simulations. The ResNet we use in experiments is a three-layer net that contains 4096 neurons and a ReLU activation function in each layer. The CNN for MNIST we use in experiments is the same architecture as that in https://github.com/pytorch/examples/blob/master/mnist/main.py.

**Weights.** As discussed in Section 3, parameter initialization is important for local elasticity based clustering methods. We find that the optimal setting always outperforms the random setting. It

| Primary Examples | 5 vs 8 | 5 vs 8 | 5 vs 8 | 5 vs 8 | 5 vs 8 | 5 vs 8 |
|---|---|---|---|---|---|---|
| Auxiliary Examples | 3, 9 | 6, 9 | 2, 3 | 2, 6 | 3 | 9 |
| $\ell_2$-relative (ours) | 75.9 | 53.7 | 54.3 | 51.2 | 53.9 | 60.6 |
| $\ell_2$-kernelized (ours) | 71.0 | 54.4 | 50.2 | 51.6 | 50.6 | 54.4 |
| BCE-relative (ours) | 50.2 | 50.4 | 50.1 | 50.2 | 52.3 | 50.1 |
| BCE-kernelized (ours) | 52.1 | 52.7 | 51.6 | 52.4 | 50.3 | 50.3 |

Table 6: Comparison of different auxiliary examples for 5 vs 8 on MNIST.

| Primary Examples | 5 vs 8 | 4 vs 9 | 7 vs 9 | 5 vs 9 | 3 vs 5 | 3 vs 8 |
|---|---|---|---|---|---|---|
| Auxiliary Examples | 3, 9 | 5, 7 | 4, 5 | 4, 8 | 8, 9 | 5 |
| $\ell_2$-kernelized (ours) | 71.0 | **63.8** | 64.6 | 67.8 | 71.5 | 78.8 |
| BCE-kernelized (ours) | 52.1 | 59.3 | 64.5 | 58.2 | 52.5 | 51.8 |
| $\ell_2$-normalized-kernelized (ours) | **71.3** | 62.3 | **64.8** | **68.0** | **73.0** | **79.5** |
| BCE-normalized-kernelized (ours) | 52.0 | 57.5 | 61.0 | 56.9 | 50.1 | 75.2 |

Table 7: Comparison between the kernelized similarity based methods and the normalized kernelized similarity based methods on MNIST.

| Primary Examples | 5 vs 8 | 4 vs 9 | 7 vs 9 | 5 vs 9 | 3 vs 5 | 3 vs 8 |
|---|---|---|---|---|---|---|
| Auxiliary Examples | 3, 9 | 5, 7 | 4, 5 | 4, 8 | 8, 9 | 5 |
| $\ell_2$-relative (unnormalized) | 51.5 | 58.4 | 55.9 | 53.0 | 69.9 | 75.5 |
| $\ell_2$-kernelized (unnormalized) | 51.4 | 56.1 | 55.3 | 57.2 | 62.1 | 75.1 |
| BCE-relative (unnormalized) | 50.1 | 50.3 | 50.7 | 53.2 | 50.9 | 50.1 |
| BCE-kernelized (unnormalized) | 50.6 | 56.9 | 59.6 | 54.4 | 52.2 | 50.8 |
| $\ell_2$-relative (ours) | **75.9** | 55.6 | 62.5 | **89.3** | 50.3 | 74.7 |
| $\ell_2$-kernelized (ours) | 71.0 | **63.8** | **64.6** | 67.8 | **71.5** | **78.8** |
| BCE-relative (ours) | 50.2 | 50.5 | 51.9 | 55.7 | 53.7 | 50.2 |
| BCE-kernelized (ours) | 52.1 | 59.3 | 64.5 | 58.2 | 52.5 | 51.8 |

Table 8: Comparison of the clustering algorithms without data normalization on MNIST. Note that we use data normalization by default.

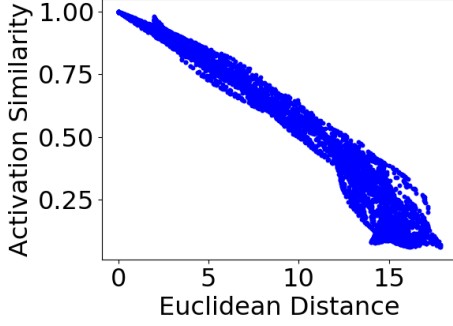
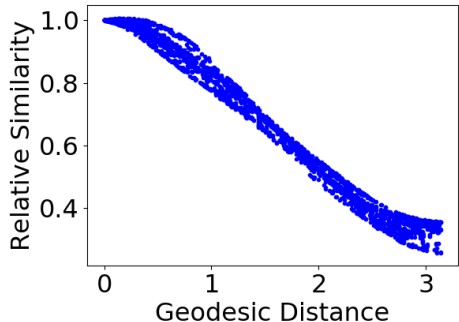

(a) Activation analysis of two-layer neural nets with ReLU activation function on the torus function.

(b) The local elasticity of two-layer neural nets with sigmoid activation function on the torus function.

Figure 7: The activation analysis and the activation function analysis of neural nets local elasticity. The Pearson correlation between the cosine similarity based on activation patterns and the corresponding Euclidean distances is $-0.97$ on the torus function. The cosine similarity within activation patterns indicates the Euclidean distance between two data points. The Pearson correlation between the relative similarity based on two-layer neural nets with sigmoid activation function and the geodesic distance is $-0.99$. It indicates that two-layer neural nets with sigmoid activation function also have a strong local elastic effect.

supports the intuition that we can learn detailed features of primary examples to better distinguish them from auxiliary examples. The results of two different types of parameter initialization are shown in Table 5.

**Auxiliary examples.**  In general, we can choose those samples that are similar to primary examples as auxiliary examples, so they can help neural nets better learn primary examples. We analyze the primary examples pair $5$ and $8$ in MNIST with $6$ different settings of auxiliary examples. We find that the choice of auxiliary examples are crucial. For example, in our case $9$ is close to $5$ and $3$ is close to $8$ based on $K$-means clustering results, so that we can choose $9$ and $3$ as auxiliary examples for models to better distinguish $5$ and $8$. The results of different auxiliary examples for $5$ vs $8$ on MNIST are shown in Table 6.

**Normalized kernelized similarity.**  The normalized kernelized similarity is

$$\overline{S}_{\mathrm{ker}}(\boldsymbol{x}, \boldsymbol{x}') := S_{\mathrm{ker}}(\boldsymbol{x}, \boldsymbol{x}')/\sqrt{S_{\mathrm{ker}}(\boldsymbol{x}, \boldsymbol{x})S_{\mathrm{ker}}(\boldsymbol{x}', \boldsymbol{x}')}.$$

The normalized kernelized similarity can achieve more stable local elasticity. We compare clustering algorithms based on kernelized similarity and normalized kernelized similarity in MNIST. We find that the performance of the normalized kernelized similarity based methods is slightly better than that of kernelized similarity based methods with the $\ell_2$ loss, although this conclusion no longer holds true with cross-entropy loss. The results of the normalized kernelized similarity are shown in Table 7.

**Expected relative change.**  We compute the average relative change of two-layer neural nets and linear neural networks fitting the torus function. The average relative change of two-layer neural nets and linear neural networks are $0.44$ and $0.68$, indicating that SGD updates of two-layer neural nets are more local than that of two-layer linear neural networks. These findings further support our local elasticity theory of neural nets.

**Activation patterns.**  The key difference between our methods and linear baselines is nonlinear activation. We analyze the relations between the patterns of ReLU activation function of two-layer neural nets and corresponding Euclidean distances. For each input $\boldsymbol{x}$, we use $\boldsymbol{a}$ to denote the binary activation pattern of each neuron. The similarity between binary activation patterns is computed from the cosine similarity function. We find that the activation similarity is linearly dependent on the Euclidean distance, meaning that closer points in the Euclidean space will have more similar activation patterns. The activation analysis of neural nets local elasticity are shown in Figure 7(a).

**Activation functions.**  We experiment on another standard activation function, sigmoid, to see its local elasticity. We find that sigmoid can also provide neural nets with the local elasticity. The local elasticity of two-layer neural nets with sigmoid activation function on the torus function is shown in Figure 7(b).

**Data normalization.**  The performance of the methods without data normalization are shown in Table 8. The results show that normalization is important for our methods, which are consistent with our analysis in Section 5.

