# OpenReview forum: "The Local Elasticity of Neural Networks"
_ICLR.cc/2020/Conference — Accept (Poster)_

### Official Review · AnonReviewer1 · 2019-10-23
**Official Blind Review #1**

**Rating:** 6

**Review:**

The paper contributes to the understanding of neural networks and provides a new clustering technique:
  1) The paper introduces the interesting notion of local elasticity which considers the relative variation in output values for two different inputs before and after an SGD-style update;
  2) The derived similarity metric between input samples (obtained by as SGD unfolds) can be used for clustering and is amenable to a kernelized formulation;
  3) Empirical measurements on visual classification tasks show that the new similarity metric can offer a better clustering performance than PCA + K-means.

I found the paper very interesting but the writing appeared somewhat unclear at times. I believe that some rewriting is needed for the authors to argue that the newly introduced elasticity metric provides a significantly new understanding of neural networks. In particular, I did not find the argumentation around explaining generalization to be very convincing or clear.

The kernelized formulation of the elasticity metric seems compelling and I found that turning the insights developed by the theoretical section of the paper into an actionable algorithm for clustering was a nice contribution.

Unfortunately, the empirical results did not really convince me that the resulting clustering algorithm really improves on the SOTA in clustering as only relatively weak baselines were considered.

I believe that considering more solid baselines for the clustering experiments would help.

**Experience Assessment:**

I have read many papers in this area.

**Review Assessment: Checking Correctness Of Derivations And Theory:**

I assessed the sensibility of the derivations and theory.

**Review Assessment: Checking Correctness Of Experiments:**

I assessed the sensibility of the experiments.

**Review Assessment: Thoroughness In Paper Reading:**

I read the paper at least twice and used my best judgement in assessing the paper.

---

> ### Author Response · Authors · 2019-11-07
> **Official Comment**
>
> Thanks for your insightful comments.
>
> The goal of this paper is to confirm the local elasticity which we had hypothesized for years. We basically took three steps to fulfill the goal. First, we used synthetic examples to confirm this phenomenon in a concrete fashion. Next, we made a connection to the theory of neural tangent kernels to corroborate our hypothesis from a theoretical perspective. Last, we introduced a clustering algorithm that leverages local elasticity. If local elasticity were not true, the performance of the algorithm would basically be the same as randomly tossing a coin. Fortunately, the algorithm performs reasonably well and thus local elasticity must hold.
>
> Therefore, the clustering algorithm was “only” to confirm the local elasticity phenomenon. In this paper we don’t recommend the use of this algorithm in practice and we even didn’t manage to further improve the efficiency of the algorithm. In fact, it would be quite straightforward to improve the algorithm by replacing the k-means with some more sophisticated algorithms, as mentioned in Section 4.2.
>
> Below please find the response in a point-by-point manner.
>
> “I found the paper very interesting but the writing appeared somewhat unclear at times. …. In particular, I did not find the argumentation around explaining generalization to be very convincing or clear.”
>
> Response: Thanks for your interest! We’ve significantly improved the writing of the paper and added more experiments, as shown in the revised version. In particular, the revision confirmed local elasticity of ResNet-152 on ImageNet. The implication of local elasticity on generalization was interpreted through stability, by making use of the intimate connection between the two established in the seminal paper by Bousquet and Elisseeff. In the revision, we elaborated on stability in a more detailed way.
>
> “Unfortunately, the empirical results did not really convince me that the resulting clustering algorithm really improves on the SOTA in clustering as only relatively weak baselines were considered. I believe that considering more solid baselines for the clustering experiments would help.”
>
> Response: We agree that our clustering algorithm at the present form does not improve on SOTA. However, as pointed out earlier, this is not the aim of the paper. As declared in the introduction, the clustering algorithm is designed to corroborate our hypothesis that neural networks (with non-linear activation) are locally elastic. Because the geodesic distance in real data is generally unknown, we cannot directly verify our hypothesis. We instead use the clustering algorithm to indirectly corroborate our hypothesis.

---

> > ### Comment · AnonReviewer1 · 2019-11-13
> > **Changing rating to weak accept**
> >
> > The authors have addressed some of my comments.
> >
> > I do believe that the submission is now stronger and change my rating to weak accept.

---

### Official Review · AnonReviewer2 · 2019-10-23
**Official Blind Review #2**

**Rating:** 6

**Review:**


[Summary]
This paper proposes and studies the “local elasticity”, a quantitative measure for the ability of neural networks to only locally change its prediction (around x) after a stochastic gradient step at x. The paper verifies experimentally that nonlinear neural nets are locally elastic through showing that an elasticity-motivated similarity score can perform clustering well.

[Pros]
The notion of local elasticity is interesting and has the potential of opening up lots of further directions. The way I understand it is to relate to memorization (as the authors have indeed discussed) --- I think “local elasticity” can be viewed as some sort of “local memorization ability”, in that the NN is able to change its prediction only in a small neighborhood of x---without affecting predictions at other remote x’s---after one SGD step on x. Conceptually this is something not covered by the existing narratives in deep learning theory, yet the phenomenon itself is quite convincing and could provide a new perspective into lots of things.

[Cons]
It feels like the experimental results in the present paper is a rather indirect evidence for the local elasticity -- that the similarity score coming from the elasticity works well for a downstream clustering task. Could there be some more direct evidence about the local elasticity? How would the elasticity compare on different architectures? In the present form the experiments perhaps at most says that the similarity score makes sense, not yet that a fully quantitative characterization of the local elasticity.

I’m also a little bit concerned about the fairness of the clustering experiment, in that the elasticity-motivated clustering algorithm utilizes an auxiliary dataset whereas simple baselines such as K-means and PCA K-means are not able to use that. Is there a way of modifying the K-means and PCA K-means so that they can also use this auxiliary dataset while still giving a sensible algorithm for the primary 2-class clustering task?


**Experience Assessment:**

I have published one or two papers in this area.

**Review Assessment: Checking Correctness Of Derivations And Theory:**

I assessed the sensibility of the derivations and theory.

**Review Assessment: Checking Correctness Of Experiments:**

I carefully checked the experiments.

**Review Assessment: Thoroughness In Paper Reading:**

I read the paper thoroughly.

---

> ### Author Response · Authors · 2019-11-07
> **Official Comment**
>
> Thanks for your insightful comments. In particular, your remark on the connection between local elasticity and the local memorization ability is very interesting and deserves future investigations.
>
> Below please find the response in a point-by-point manner.
>
> “It feels like the experimental results in the present paper is a rather indirect evidence for the local elasticity … In the present form the experiments perhaps at most says that the similarity score makes sense, not yet that a fully quantitative characterization of the local elasticity.”
>
> Response: The paper first provided direct evidence of local elasticity using synthetic data in Figure 1, in which we can get the exact geodesic distance between feature vectors. In the updated version, we further provide direct evidence using examples from real data in Appendix A.2.  Using pre-trained ResNet-152 on ImageNet, specifically, we found that the change of the predictions (KL divergence of 0.03) on the tiger cat (Figure 6(b)) is more drastic than that (KL divergence of 0.002) on the warplane (Figure 6(c)) after a SGD update on the tabby cat (Figure 6(a)), although the Euclidean distance (511) between the warplane and the tabby cat is smaller than that (854) between the tiger cat and the tabby cat.
>
> We acknowledge that the evidence provided by the clustering algorithm is indirect. However, it is quite difficult to directly confirm local elasticity on real data since the geodesic distance is generally unknown. In fact, it took us a very long time to devise the clustering algorithm for indirect evidence. Moreover, if local elasticity were not true, it would be very challenging to explain the effectiveness of the clustering algorithm. As such, we believe the indirect evidence is quite solid. As for different architectures, the paper actually already compared three architectures, FNN, CNN, and ResNet, in Table 3. We found that CNN shows very pronounced local elasticity. We can explore more architectures as needed during the rebuttal period.
>
> “I’m also a little bit concerned about the fairness of the clustering experiment, Is there a way of modifying the K-means and PCA K-means so that they can also use this auxiliary dataset while still giving a sensible algorithm for the primary 2-class clustering task?”
>
> Response: Thanks for making this point. In fact, it is entirely nontrivial to make use of the auxiliary dataset for K-means and PCA K-means, if provided. Our goal is not to design an effective clustering algorithm for practical use. Instead, it is to corroborate our hypothesis that neural networks (with non-linear activation) are locally elastic. So the comparison is to only show that the clustering algorithm effectively leverages local elasticity. In the revision, we’ve made this point clear.

---

### Official Review · AnonReviewer3 · 2019-10-24
**Official Blind Review #3**

**Rating:** 6

**Review:**

This paper studies an interesting phenomenon in neural network models that the classifier's prediction at a one input will not be significantly perturbed after the classifier is updated via sgd at another input that is dissimilar from the former one. This phenomenon is termed as the local elasticity, which provides another perspective seeking to interpret the neural networks. They present that this local elasticity characteristic does not hold for linear models. To further investigate this property, the paper introduces the relative similarity and kernelized similarity based on which a k-means like clustering algorithm is developed to further find fine-grained clusters within a coarse-grained category, like dogs and cats from the mammal category. Interpreting neural networks is a hot research topic, and a paper advancing knowledge in this area is certainly welcome. The paper is well presented (with a small typo in the definition of S_ker(x,x)). In the experiments, it will be interesting to further investigate how the local elastic property changes with large batch size in that large batch size may encourage more diversity of the examples in a batch. Furthermore, it will be even more interesting to explore how these similarities can improve the performance of a simple k-nearest neighbor classifier.

**Experience Assessment:**

I have read many papers in this area.

**Review Assessment: Checking Correctness Of Derivations And Theory:**

I assessed the sensibility of the derivations and theory.

**Review Assessment: Checking Correctness Of Experiments:**

I carefully checked the experiments.

**Review Assessment: Thoroughness In Paper Reading:**

I read the paper thoroughly.

---

> ### Author Response · Authors · 2019-11-07
> **Official Comment**
>
> Thanks for your insightful comments. Below please find the response in a point-by-point manner.
>
> “The paper is well presented (with a small typo in the definition of S_ker(x,x)).”
>
> Response: Thanks for pointing out these. We’ve fixed these typos in our updated version.
>
> “In the experiments, it will be interesting to further investigate how the local elastic property changes with large batch size in that large batch size may encourage more diversity of the examples in a batch.”
>
> Response: It’s a very interesting direction for future work. In the case of using mini-batch SGD, the prediction change of a point is presumably a weighted effect from all points in the mini-batch. It requires quite an amount of effort to investigate this extension, and we are working on it now and will report any findings once available. Having said this, the aim of this paper is to identify the local elasticity phenomenon and we believe our setting, albeit simple, is sufficient for the confirmation of this phenomenon.
>
> “Furthermore, it will be even more interesting to explore how these similarities can improve the performance of a simple k-nearest neighbor classifier.”
>
> Response: Thanks for pointing out this good direction for the application of our local elasticity! K-NN is an effective algorithm if we have a good distance measure and sufficient data points. This paper mainly focuses on corroborating our hypothesis that neural networks (with non-linear activation) are locally elastic. For any test point, it is possible to leverage the local elasticity phenomenon to evaluate the “distance” between this point and any other point. With the distance information in place, we can apply K-NN to predict the label of the test point. In turn, we believe that this use further illustrates the power and profound implication of local elasticity.

---

> ### Author Response · Authors · 2019-11-13
> **Thanks for suggesting the investigation of local elasticity with mini-batch SGD**
>
> In Appendix A.2 of the revision, we added a simulation study showing that the local elasticity phenomenon persists with mini-batch SGD. In short, a test image has a large change in its prediction after an SGD update if it is similar to at least one of the images in the mini-batch used for computing the gradient.

---

### Author Response · Authors · 2019-11-13
**One more simulation study added**

The revision included an additional simulation showing the presence of local elasticity in the case of mini-batch SGD. Details are given in Appendix A.2.

---

### Decision · Program_Chairs · 2019-12-19

**Decision:**

Accept (Poster)

**Comment:**

This paper presents a new phenomenon referred to as the "local elasticity of neural networks". The main argument is that the SGD update for nonlinear network at a local input x does not change the predictions at a different input x' (see Fig. 2). This is then connected to similarity using nearest-neighbor and kernel methods. An algorithm is also presented.

The reviewers find the paper intriguing and believe that this could be interesting for the community. After the rebuttal period, one of the reviewers increased their score.

I do agree with the view of the reviewers, although I found that the paper's presentation can be improved. For example, Fig. 1 is not clear at all, and the related work section basically talks about many existing works but does not discuss why they are related to this work and how this work add value to this existing works. I found Fig. 2 very clear and informative. I hope that the authors could further improve the presentation. This should help in improving the impact of the paper.

With the reviewers score, I recommend to accept this paper, and encourage the authors to improve the presentation of the paper.